# Sub-volt high-speed silicon MOSCAP microring modulator driven by high-mobility conductive oxide

Wei-Che Hsu[1,2], Nabila Nujhat[1], Benjamin Kupp [1], John F. Conley Jr [1], Haisheng Rong [3], Ranjeet Kumar [3] & Alan X. Wang [1,2] ✉

Silicon microring modulator plays a critical role in energy-efficient optical interconnect and optical computing owing to its ultra-compact footprint and capability for on-chip wavelength-division multiplexing. However, existing silicon microring modulators usually require more than 2 V of driving voltage ($V_{pp}$), which is limited by both material properties and device structures. Here, we present a metal-oxide-semiconductor capacitor microring modulator through heterogeneous integration between silicon photonics and titanium-doped indium oxide, which is a high-mobility transparent conductive oxide (TCO) with a strong plasma dispersion effect. The device is co-fabricated by Intel's photonics fab and our in-house TCO patterning processes, which exhibits a high modulation efficiency of 117 pm/V and consequently can be driven by a very low $V_{pp}$ of 0.8 V. At a 11 GHz modulation bandwidth where the modulator is limited by the RC bandwidth, we obtained 25 Gb/s clear eye diagrams with energy efficiency of 53 fJ/bit.

Optical microring resonators have emerged as a key building block of photonic integrated circuits (PICs) that can function as versatile optical devices, including modulators, wavelength filters and multiplexers, in comb lasers, weight banks for neuromorphic computing, and optical sensors[1–3]. They are playing increasingly critical roles in optical communication, optical interconnects, optical computing, and biomedical sensing due to their ultra-compact footprint and capability for on-chip wavelength-division multiplexing (WDM)[4]. Although optical microring resonators have been implemented on various platforms such as thin-film $LiNbO_3$[5], silicon nitride[6], and plasmonics[7], active silicon microring modulators (Si-MRMs) that can perform high-speed electro-optic (E-O) modulation as the photonic engine for future PICs remains as the climax of research[8–10]. Silicon photonics provides a mature, cost-effective platform to integrate Si-MRMs with lasers, photodetectors, passive optical devices, and even microelectronic circuits through standard foundry fabrication, which is still the only feasible solution to build large-scale PICs with both active and passive devices for complex systems[11,12].

Existing Si-MRMs available in the foundry process are based on reversed PN junctions, which can achieve ultra-high modulation bandwidth but usually require more than 2 V of driving voltage ($V_{pp}$)[13–16]. Such high $V_{pp}$ is induced by the relatively weak plasma dispersion effect of silicon and the small capacitance per unit length of the reversed PN-junction, which is usually less than 2 fF/μm[17,18]. The high $V_{pp}$ makes it unfeasible to drive Si-MRMs directly by CMOS logic circuits. For example, today's 5 nm CMOS has a core supply of 0.65 V and input/output supply of 1 V[19]. Therefore, high voltage-swing CMOS transmitter (TX) circuits that consume hundreds of milliwatts of power must be used to drive the Si-MRMs, and the energy consumption of the Si-MRM is insignificant compared with that of its CMOS driver[20]. For instance, the 106 Gb/s 2.5 $V_{pp}$ Si-MRM driver using 28 nm CMOS process consumes 1.33 pJ/bit energy with Pulse-amplitude modulation 4-level (PAM-4) while the Si-MRM itself usually consumes less than 100 fJ/bit energy[21]. Therefore, reducing the driving voltage of the modulator not only lowers the power consumption of the modulator itself, but also enables energy-efficient and high-speed modulator drivers using advanced CMOS nodes that cannot support the high driving

[1]School of Electrical Engineering and Computer Science, Oregon State University, Corvallis, OR, USA. [2]Department of Electrical and Computer Engineering, Baylor University, Waco, TX, USA. [3]Intel Corporation, 3600 Juliette Ln, Santa Clara, CA, USA. ✉e-mail: alan_wang@baylor.edu

voltage of existing silicon photonic modulators[22]. As a comparison, MOSCAP-driven Si-MRMs can achieve a much larger capacitance density using ultra-thin high dielectric constant insulators such as HfO₂. In addition, MOSCAP devices allow heterogeneous integration of more E-O efficient gate materials with Si waveguide such as III-V compound semiconductors and graphene[23–30]. In the past two decades, high-speed MOSCAP Si-MRMs, including heterogeneously integrated functional materials, have been demonstrated (Supplementary Information I)[7,10,31–37]. Nevertheless, there is still a strong desire to develop high-speed Si-MRMs with sub-volt $V_{pp}$ and large E-O modulation efficiency.

In this article, we demonstrate a highly efficient MOSCAP Si-MRM heterogeneously integrated with titanium-doped indium oxide (ITiO), a TCO with high carrier mobility. The focus of this study is to achieve sub-volt $V_{pp}$ modulation with a high E-O bandwidth, which requires optimization of the quality factor (Q-factor) and the E-O efficiency of the MRM. A higher Q-factor allows for narrower resonant spectra, enabling lower $V_{pp}$. However, it also leads to a longer photon lifetime, which imposes a limitation on the modulation bandwidth. Therefore, enhancing the E-O efficiency of the MRM through other mechanisms using more efficient gate material and higher capacitance density becomes crucial. The ITiO-gated MOSCAP Si-MRM in this work improved the E-O efficiency by narrowing the microring waveguide width, which effectively improved the overlapping factor between the accumulated carriers and the optical mode profile[38]. The utilization of high-mobility ITiO reduces the optical waveguide absorption, enabling a balanced Q-factor for sub-volt $V_{pp}$ modulation while still supporting a large photon lifetime-limited bandwidth. Additionally, the high-mobility ITiO, along with optimized doping on the Si microring waveguide and metal electrode patterning, improved the RC bandwidth significantly compared with all previous work[39,40]. As a result, the ITiO-gated MOSCAP Si-MRM can be driven by 0.8 $V_{pp}$ with 11 GHz E-O bandwidth, demonstrating a clear eye diagram at 25 Gb/s with an energy efficiency of 53 fJ/bit. In addition, we would like to point out that TCOs have emerged as a new E-O modulation material for integrated photonics in recent years[41]. Although high bandwidth was claimed[42,43], these devices only demonstrated small-signal RF modulation and no clear eye diagrams have been reported above 5 Gb/s.

In this work, the ITiO-gated MOSCAP Si-MRM represents the first TCO modulator capable of high-speed E-O modulation with clear eye diagrams exceeding 25 Gb/s, which marks a significant milestone in the development of TCO modulators. Through further optimization of the device structure to enhance the RC bandwidth, which is the primary constraint on our current device, we propose a high-mobility TCO to enable a much higher bandwidth while operating at even lower driving voltages. This advancement holds the potential for high-speed E-O modulation with sub-volt driving voltages, thereby paving the way for future advancements in energy-efficient optical communication and computation systems.

## Results
### Device design
Figure 1 illustrates the design of the ITiO-gated MOSCAP Si-MRM, including a 3D schematic diagram and a cross-sectional view of the active region. The device comprises a 300 nm thick Si rib waveguide with a 100 nm slab thickness. A waveguide width of 300 nm is selected to enhance the E-O efficiency, while a radius of 8 μm is chosen to reduce the bending loss (Supplementary Information II). The Si doping profile of p ($1 \times 10^{17}$ cm⁻³), p+ ($3 \times 10^{18}$ cm⁻³), and p ++ ($1 \times 10^{20}$ cm⁻³) are designed to reduce the series resistance without compromising the optical absorption in the Si waveguide. The entire device has a background Si p-doping, and the p+ region covers the top of the ring

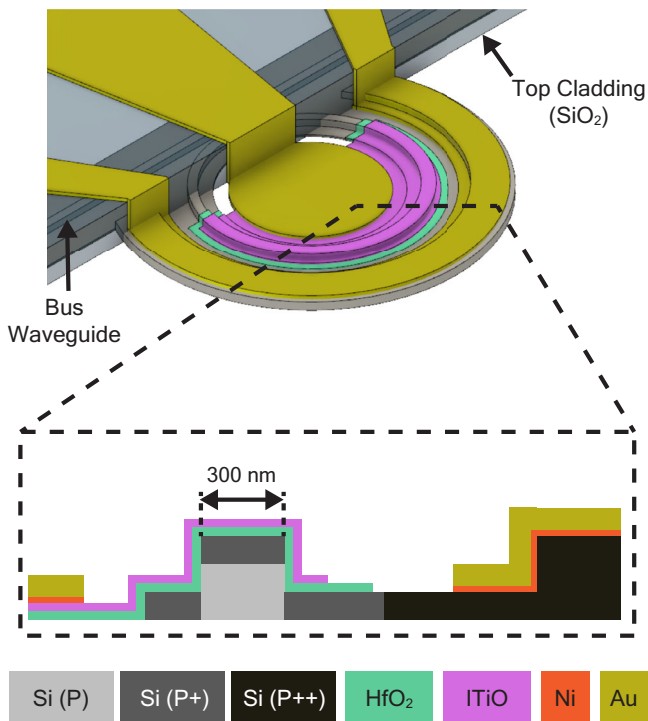

**Fig. 1 | 3D schematic diagram of an ITiO-gated MOSCAP Si-MRM.** The dashed line box shows the cross-sectional view in the active region of the ITiO-gated MOSCAP Si-MRM.

waveguide and part of the Si slab. The p ++ region is placed 600 nm away from the ring waveguide to maintain the Q-factor. Such a doping design allows the passive Si microring resonator to achieve a high Q-factor of 20,000 near the critical coupling condition. The active E-O modulation region, which represents approximately 62.5% of the microring circumference (L = 31.4 μm), consists of a 10 nm thick HfO₂ insulator layer and a 14 nm ITiO layer on the top. The Ni/Au electrodes traverse the bus waveguide via the top SiO₂ waveguide cladding, forming ohmic contacts with the ITiO gate and the Si substrate of the MOSCAP.

Figure 2a shows the simulated carrier density distribution in the cross-sectional waveguide of the ITiO-gated MOSCAP Si-MRM in the active region. When a negative bias is applied, holes and electrons accumulate at the Si/HfO₂ and HfO₂/ITiO interfaces, respectively. Accumulated electrons within the ITiO layer is less than 1 nm thick, which is significantly thinner than the hole accumulation layer in the Si waveguide due to the different Debye lengths caused by the varying carrier concentration and dielectric constant[44]. Figure 2b presents the optical mode profile of the transverse-electric (TE) mode in the ITiO-gated MOSCAP Si-MRM. Due to the bending ring waveguide, the optical mode profile shifts toward the outer sidewall of the waveguide. Upon a negative bias, the optical mode interacts with the accumulation charges in both ITiO and Si, enabling E-O modulation and blue-shifting the resonant wavelength ($\Delta\lambda_{res}$). Our previous studies have discussed that phase modulation with moderate index change is majorly determined by the total charge variation, allowing us to simplify the simulation model to a uniform distribution with 1 nm thick accumulation layers, aligning well with experimental results[45,46]. Therefore, Fig. 2b also provides a zoomed-in view of the optical mode profile at the HfO₂/ITiO interfaces, showcasing examples at 0 V and −1 V biases with a 1 nm thick uniform accumulation layer. Figure 2c shows the simulated transmission spectra under different gate biases. Our simulation assumes a dielectric constant of 12 for HfO₂ and carrier mobility

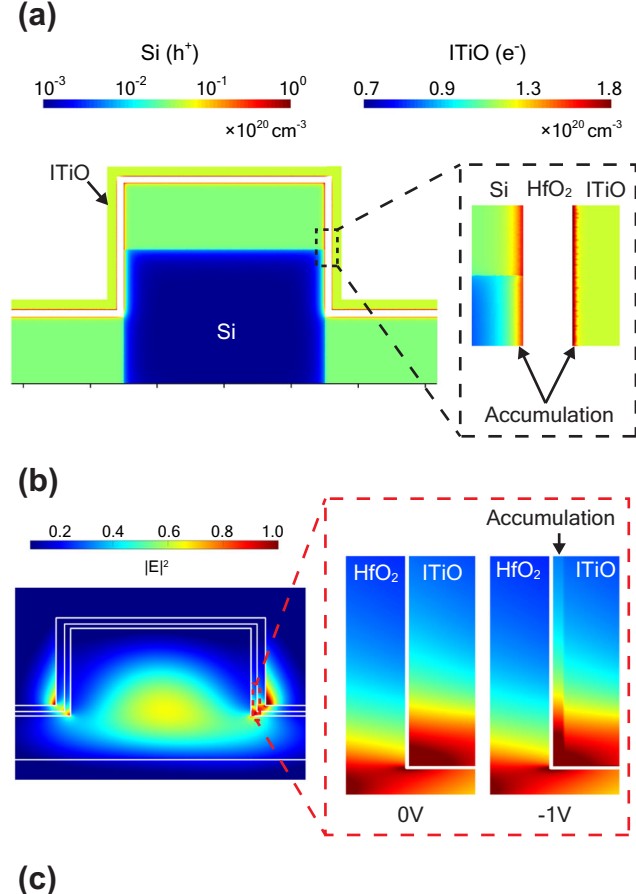

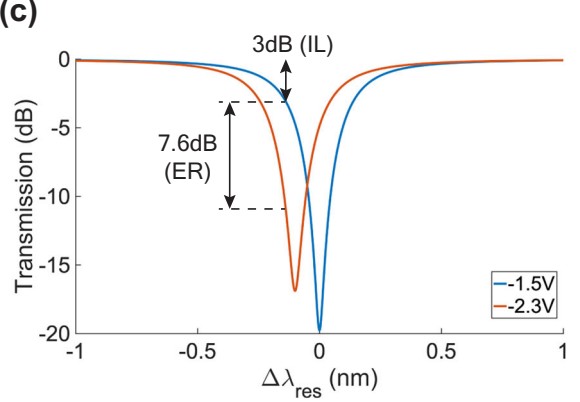

**Fig. 2 | Simulation analysis of ITiO-Gated MOSCAP Si-MRM characteristics.**
**a** Simulated carrier distribution in the cross-sectional waveguide of the ITiO-gated MOSCAP Si-MRM at a bias of −2.3 V. Negative bias on ITiO gate results in hole accumulation at the Si/HfO₂ interface and electron accumulation at the HfO₂/ITiO interface. Note: the color bars representing carrier concentrations for Si and ITiO are in different scales. **b** Optical mode profile of the TE mode in the ITiO-gated MOSCAP Si-MRM waveguide. Zoomed-in views at the HfO₂/ITiO interfaces at 0 V and −1 V biases. **c** Static simulation of the blue-shifted spectra under −1.5 V and −2.3 V.

---

of 62 cm²/(V·s) with a concentration of $1.2 \times 10^{20}$ cm⁻³ for ITiO, chosen to align with the experimental material properties (Supplementary Information III, IV). Using such parameters, the ITiO-gated MOSCAP Si-MRM achieves a high Q-factor of 5000 at 0 V with an E-O efficiency of 125 pm/V in the accumulation mode. To balance the output optical power and the driving voltage, we chose a 3 dB insertion loss (IL) point based on the notable reduction in the Lorentzian shape of the microring transmission. In addition, an extinction ratio (ER) at 6 dB is adopted to enable clear eye diagrams, especially when transmitting high-speed data.

The combination of 3 dB IL and 6 dB ER is also recommended as a practical and effective choice for E-O modulation in optical transmitters[47]. The simulation results in Fig. 2c also demonstrate that the ITiO-gated MOSCAP Si-MRM can achieve a 7.6 dB ER with 3 dB IL using only 0.8 $V_{pp}$, highlighting the potential for efficient modulation.

Figure 3a presents the scanning electron microscope (SEM) image of the passive Si microring resonator, which was fabricated by Intel's 300-millimeter silicon-on-insulator (SOI) photonics process. To complete the entire process of the active ITiO-gated MOSCAP Si-MRM, we continued the fabrication using the clean-room facility at Oregon State University. The SiO₂ top cladding in the active region was then etched by reactive ion etching (RIE). Despite the high etching selectivity between SiO₂ and Si, some Si waveguide was still inadvertently etched during the process. As a result, the width of the Si microring waveguide was slightly narrower than the designed value, measuring 290 nm, as depicted in Fig. 3b. After the entire fabrication process, Fig. 3c shows a top view of the fabricated ITiO-gated MOSCAP Si-MRM. Additionally, Fig. 3d provides a zoomed-in SEM image highlighting the ITiO region, represented by a false purple color.

## DC characterization

The ITiO-gated MOSCAP Si-MRM was designed to operate at O-band. The normalized transmission spectra with various gate voltages are shown in Fig. 4a. The device was designed near critical coupling, resulting in a deep resonant dip to enable a modulation condition with an ER exceeding 6 dB and an IL of 3 dB. In addition, the critical coupling condition plays an important role in achieving sub-volt modulation. Critical coupling leads to a sharp and deep resonance, which minimizes the required driving voltage as discussed in Supplementary Information V. Applying a negative bias to the ITiO gate causes the accumulated carriers in both the Si waveguide and ITiO gate, resulting in $\Delta\lambda_{res}$. Simultaneously, such accumulated charges also increase optical absorption and decrease the Q-factor. The measured Q-factors and $\Delta\lambda_{res}$ are plotted in Fig. 4b. The ITiO-gated MOSCAP Si-MRM exhibits a Q-factor of approximately 4600 at 0 V, supporting an optical bandwidth of 50 GHz. The Q-factor is used to determine the optical loss of the ITiO-gated MOSCAP waveguide at 138 dB/cm. As a comparison, the passive silicon waveguide microring exhibited a loss of 24 dB/cm before the ITiO deposition. Therefore, the introduction of ITiO leads to a significant increase of the optical loss and is a critical factor for device design. For gate biases ranging from 0 V to −1.5 V, the ITiO-gated MOSCAP Si-MRM operates in the depletion mode due to the non-ideal flat-band voltage ($V_{FB}$), achieving an E-O efficiency of 87 pm/V. Once the negative bias exceeds −1.5 V, the MOSCAP transitions into the accumulation mode, and $\Delta\lambda_{res}$ becomes more linear, leading to a higher E-O efficiency of 117 pm/V, corresponding to a very low $V_\pi$·L of 0.12 V·cm. Therefore, the ITiO-gated MOSCAP Si-MRM operates in the accumulation mode to achieve a low driving voltage ($V_{pp}$). Additionally, it is worth noting that the experimental Q-factor (4600) and E-O efficiency (117 pm/V) obtained were only slightly lower than the simulated values of Q-factor (5000) and E-O efficiency (125 pm/V) with an acceptable error margin of 8%. The modulation wavelength ($\lambda_{MOD}$) was fine-tuned by a tunable laser to ensure an IL of 3 dB at −1.5 V. The optical transmission for different gate biases at $\lambda_{MOD}$ is shown in Fig. 4c. An ER of 6 dB and an IL of 3 dB can be achieved with a bias voltage of −1.9 V and a voltage swing of 0.8 $V_{pp}$ (−1.5 V - −2.3 V). The observed ER, though slightly lower than the expected value, can be attributed to the slightly lower experimental Q-factor and E-O efficiency.

## High-speed characterization

The E-O response ($S_{21}$) of the ITiO-gated MOSCAP Si-MRM was measured under the condition of 3 dB IL at −1.5 V. By varying the frequency

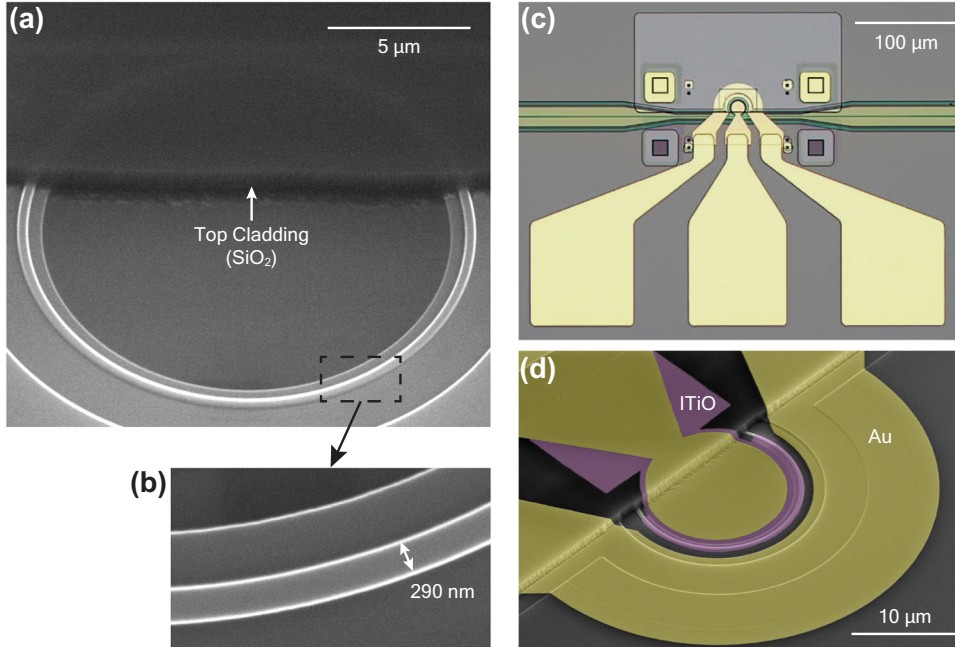

**Fig. 3 | Fabricated ITiO-gated MOSCAP Si-MRM. a** SEM image of the passive Si microring resonator fabricated by Intel's photonics fab, showing the etched SiO$_2$ top cladding in the active region after RIE. **b** Top-view SEM image highlighting the narrow Si microring waveguide. **c** Optical image of the fabricated ITiO-gated MOSCAP Si-MRM with high-speed Ni/Au electrodes. **d** SEM image of the fabricated ITiO-gated MOSCAP Si-MRM with false colors.

of the input sine wave driving signal, the corresponding output RF power was measured by a microwave spectrum analyzer (MSA). Figure 5a shows an example of the output signal on the MSA at 10 GHz. Since the output signals were amplified by a 42 GHz photodetector with a built-in transimpedance amplifier, the output RF power was normalized to the RF power at a low frequency of 500 MHz. The measured and normalized data were plotted in Fig. 5b, with the fitting curve represented by an orange dashed line. The E-O response exhibits the peaking effect[48], enhancing the 3 dB bandwidth to 11 GHz. It provides the potential for supporting the ITiO-gated MOSCAP Si-MRM in achieving non-return-to-zero (NRZ) modulation at data rates exceeding 20 Gb/s.

For the E-O modulation measurement, the ITiO-gated MOSCAP Si-MRM was biased at the IL of 3 dB at −1.5 V and driven by 0.8 V$_{pp}$ NRZ pseudorandom binary sequence (PRBS) signals. PRBS9 was used for data rates lower than 10 Gb/s, and PRBS15 was employed for higher data rates to ensure sufficient data randomness for characterization. The device's capacitance was measured to estimate the modulation energy consumption using CV$^2$/4. It is worth noting that the measured capacitance included parasitic capacitance arising from the waveguide slab. This parasitic capacitance does not play a role in modulation as discussed in Supplementary Information IV. To accurately estimate the best possible modulation energy, it is crucial to exclude such effect. Since the capacitance changes with the applied gate bias as indicated in the C-V characteristics (Supplementary Information IV), the capacitance (after eliminating the parasitic capacitance) was found to be 333 fF at −1.9 V, which corresponds to the center of the 0.8 V voltage swing (−1.5 V to −2.3 V). Considering the modulation energy consumption (CV$^2$/4), the estimated energy consumption is 53 fJ/bit. Figure 6a presents the obtained optical eye diagrams at different data rates using a digital communication analyzer (DCA). It is evident from the diagrams that the eye remains open even at the data rate of 25 Gb/s. To push for even higher data rates, the S$_{21}$ data from Fig. 5b was utilized as the input into the arbitrary waveform generator (AWG) to pre-emphasize the signals to enhance the quality of received signals at the DCA. Additionally, the ITiO-gated MOSCAP

Si-MRM was driven with a higher voltage. Figure 6b shows the optical eye diagrams obtained through this approach, incorporating the pre-emphasis signals and a driving voltage of 1.75 V$_{pp}$. Pre-emphasis can lead to a cleaner and more open eye diagrams. However, it's important to note that pre-emphasis doesn't increase the actual bandwidth. Compared to Fig. 6a, it becomes apparent that the open eye at 25 Gb/s is even clearer, and the eyes are successfully opened up to 35 Gb/s. At higher data rates, the primary limiting factor for the eye diagram is the actual bandwidth of the device. Additionally, in our testing configuration, the use of an optical amplifier introduced amplified spontaneous emission (ASE) noise, which had a detrimental effect on the clarity of the eye diagram. However, the ASE noise can be filtered out by a narrow bandpass optical filter.

### Device optimization

Figure 5b demonstrated that the ITiO-gated MOSCAP Si-MRM achieved an E-O bandwidth of 11 GHz, which is limited by the RC bandwidth since the photon lifetime-limited bandwidth supports up to 50 GHz with a Q-factor of 4600. The device's high overall capacitance of 500 fF (Supplementary Information IV) hinders the bandwidth improvement. Although higher doping concentrations of the TCO and Si materials can lower the resistance, it will introduce greater optical absorption loss that can suppress the Q-factor with the price of higher driving voltages. Consequently, the most effective approach toward enhanced bandwidth lies in reducing the overall capacitance. However, it's worth noting that a high capacitance density is crucial for improving the modulation efficiency. Therefore, optimal capacitance density with balanced performance is a crucial aspect of the MOSCAP microring modulator design.

In the current device, the MOSCAP is formed by covering the top and two sidewalls of the waveguide with ITiO, as well as the 500 nm width of the Si slab. By calculation shown in Supplementary Information VI, the outer sidewall of the ring contributes over 50% of the total modulation, while the top of the waveguide and the slab act more like parasitic capacitance with minimal contribution to E-O modulation. This insight analysis suggests

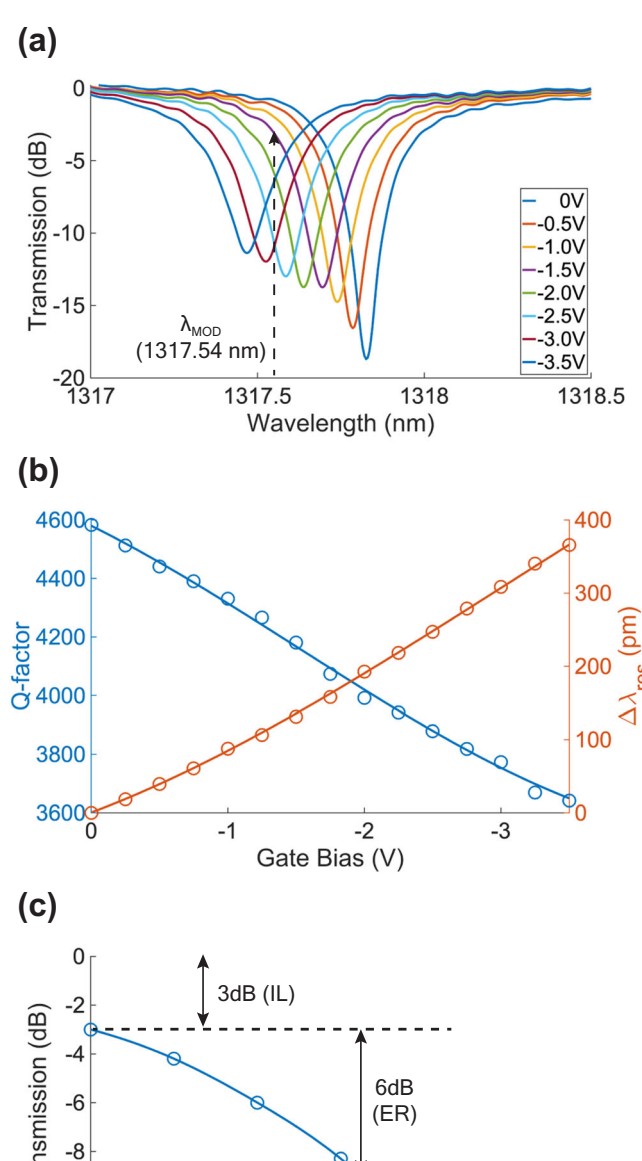

**(a)**

**(b)**

**(c)**

**Fig. 4 | DC characterization. a** Normalized transmission spectra of the ITiO-gated MOSCAP Si-MRM with different gate biases. **b** Measured Q-factor (blue) and $\Delta\lambda_{res}$ (orange) as a function of the gate bias. **c** Transmission at $\lambda_{MOD}$ (1317.54 nm) with respect to the gate voltage. It biases at −1.9 V with 0.8 $V_{pp}$ (−1.5 V to −2.3 V) to achieve an ER of 6 dB with an IL of 3 dB.

5600 with an E-O efficiency of 148 pm/V, as depicted in Fig. 7b. Hence, it can achieve a 6 dB ER with a 3 dB IL using only 0.5 $V_{pp}$. Moreover, with this improved structure and the material properties, the total capacitance can be reduced from 500 fF to 218 fF, and the total series resistance will be 18 Ω, effectively supporting an RC bandwidth up to 40.6 GHz if we exclude the effect of source impedance. Due to the ultra-short electrode length of the microring modulator, we can simplify the device to a lumped component in our bandwidth simulation and characterization. The simulation accounts for both photon lifetime-limited bandwidth and RC bandwidth, resulting in a simulated E-O bandwidth that extends to 52 GHz, as shown in Fig. 7c. In addition, the energy efficiency can be improved to 13.6 fJ/bit. Another possibility to further improve the energy efficiency is to utilize subwavelength microring structures with minimized bending loss to achieve ultra-high overlapping factor between the optical mode and the accumulated carriers, which worth future investigations[50].

## Discussion

This work successfully demonstrated the integration between silicon photonics and high-mobility TCO material to create a highly efficient ITiO-gated MOSCAP Si-MRM. The device was co-fabricated by Intel's photonics fab and TCO patterning processes at Oregon State University. It exhibited an exceptional E-O efficiency of 117 pm/V, along with a $V_\pi$•L of 0.12 V•cm. With an E-O bandwidth of 11 GHz, it achieved a 25 Gb/s open eye at a sub-volt $V_{pp}$ of 0.8 V with an energy efficiency of 53 fJ/bit. With further optimization of the device structure, it has the potential to achieve a 0.5 $V_{pp}$ while increasing the E-O bandwidth to 52 GHz, enabling data encoding at 100 Gb/s for the next generation of high-speed optical communication with an energy efficiency of 13.6 fJ/bit. In conclusion, the highly efficient ITiO-gated MOSCAP Si-MRM presented in this work has a significant impact on energy-efficient optical communication and computation. Its sub-volt driving voltage offers the feasibility of direct CMOS driving without any voltage amplifier, which can potentially reduce the transmitter-side power consumption by an order of magnitude. It will also bridge the gap of the driving voltage between neuromorphic computing and photonic modulators, enabling low-energy photonic computing for artificial intelligence.

## Methods
### Fabrication

The passive Si microring resonator was fabricated on a SOI wafer using Intel's photonics fab. The Si microring has a narrow waveguide width of 300 nm and a waveguide height of 300 nm while leaving a 100 nm thick Si slab to ensure good optical mode confinement in the Si waveguide and proper electrical conduction. To create the MOSCAP on the microring, the top $SiO_2$ cladding in the active region of the microring was selectively patterned using regular photolithography, followed by RIE, as shown in Fig. 3(a). Next, a 10 nm $HfO_2$ insulator layer was deposited on the entire SOI substrate by atomic layer deposition (ALD) using tertrakis (eythylmetylamido)-Hf (TEMA) and $H_2O$ at 300 °C. Subsequently, a 14 nm layer ITiO was RF-sputtered onto the $HfO_2$ layer at a high substrate heating temperature of 500 °C, which covered the entire wafer with ITiO. The ITiO layer in the active region was patterned using a two-step process. Electron beam lithography (EBL) with RIE was employed to accurately define the desired ITiO pattern within the active region. Subsequently, regular photolithography with wet etching (ITO etchant) was used to remove any residual ITiO. After these two steps, the ITiO layer only covered approximately 62.5% of the microring circumference. Prior to the metal deposition, the $HfO_2$ layer in the Si contact region

that the total capacitance can be significantly reduced by covering only the outer sidewall of the ring, as illustrated in Fig. 7a. In parallel, the thickness of $HfO_2$ is reduced from 10 nm to 6 nm, enhancing the capacitance density and improving E-O efficiency. Furthermore, employing a higher mobility TCO, such as hydrogen-doped indium oxide (IHO)[49], not only improves the Q-factor by reducing the optical loss but also decreases the series resistance. IHO has reported a high-mobility of 150 cm²/(V·s) with a concentration of $1.5 \times 10^{20}$ cm⁻³. Assuming the active E-O modulation region occupies 70% of the microring, this structure allows the modulator to achieve a Q-factor of 6000 at 0 V. When the device is biased at −1.5 V, the Q-factor slightly reduces to

**(a)**

**(b)**

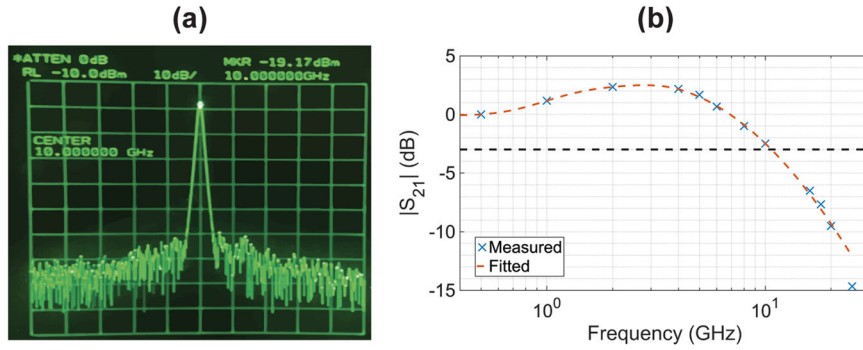

**Fig. 5 | E-O response characterization. a** Output signal on the MSA at 10 GHz. **b** Normalized E-O response ($S_{21}$) of the ITiO-gated MOSCAP Si-MRM in the frequency range of 500 MHz to 25 GHz. The fitting curve is formed by fitting the discrete data points of the measured RF power to show the device's EO bandwidth, which considers the effect of detuning, peaking, and RC bandwidth. Note: A tunable laser was fine-tuned to the input wavelength to achieve an IL of 3 dB at −1.5 V.

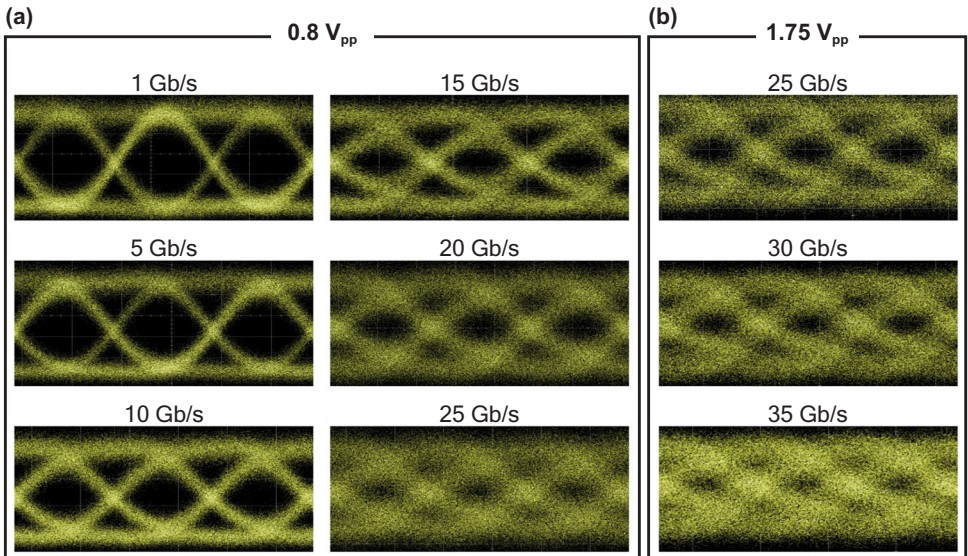

**Fig. 6 | Measured NRZ modulation eye diagrams of the ITiO-gated MOSCAP Si-MRM with different data rates. a** driving voltage of 0.8 $V_{pp}$ without any pre-emphasis signal. **b** driving voltage of 1.75 $V_{pp}$ with the pre-emphasis signal.

was patterned using EBL and removed by RIE. The Ni/Au electrical contacts were patterned on the ITiO gate and the Si substrate using EBL, followed by thermal evaporation and lift-off. These contacts were patterned approximately 1.2 µm away from the microring waveguide. Finally, the Ni/Au coplanar ground-signal-ground (GSG) electrode pads were patterned using regular photolithography followed by thermal evaporation and lift-off and were connected to the electrical contacts patterned in the previous step.

### Eye diagrams testing

Figure 8a illustrates the testing setup utilized for measuring optical eye diagrams. A PRBS electrical signal was generated using a 92 GSa/s AWG (Keysight M8196A), and it was combined with a DC bias through a 50-GHz bias tee (Keysight 11612B), ensuring optimal modulation of the electrical signal. Subsequently, this combined signal was applied to the device via an Infinity 40-GHz high-speed GSG probe. Simultaneously, a tunable laser (TL) (Santec TSL-570) was employed to generate the optical input, and the optical signals were coupled in and out through waveguide grating couplers. The modulated optical signal was then amplified by an O-Band

Praseodymium-Doped Fiber Amplifier (PDFA) (Thorlabs PDFA100) before being detected by a 65 GHz optical module (Keysight N1030A) plugged to a DCA (Keysight N1000A). Finally, the DCA enables the acquisition of the optical eye diagram.

### E-O response testing

The E-O response shown in Fig. 5b was characterized using the experimental setup depicted in Fig. 8b. The single-frequency sine wave generated by the AWG was combined with a DC voltage using the bias tee before being utilized to drive the ITiO-gated MOSCAP Si-MRM. The device was modulated by the input single-frequency sine wave, resulting in an output-modulated optical signal detected by a 42 GHz photodetector with a built-in transimpedance amplifier (Thorlabs RXM42AF). Subsequently, the detected signal was further analyzed using a 26 GHz MSA (HP8562A). To examine the frequency-dependent behavior of the E-O response, the input frequency of the sine wave was scanned, and the corresponding changes in the output power displayed on the MSA were observed. By systematically measuring the output power at different input frequencies, the $S_{21}$ response of the ITiO-gated MOSCAP Si-MRM was characterized.

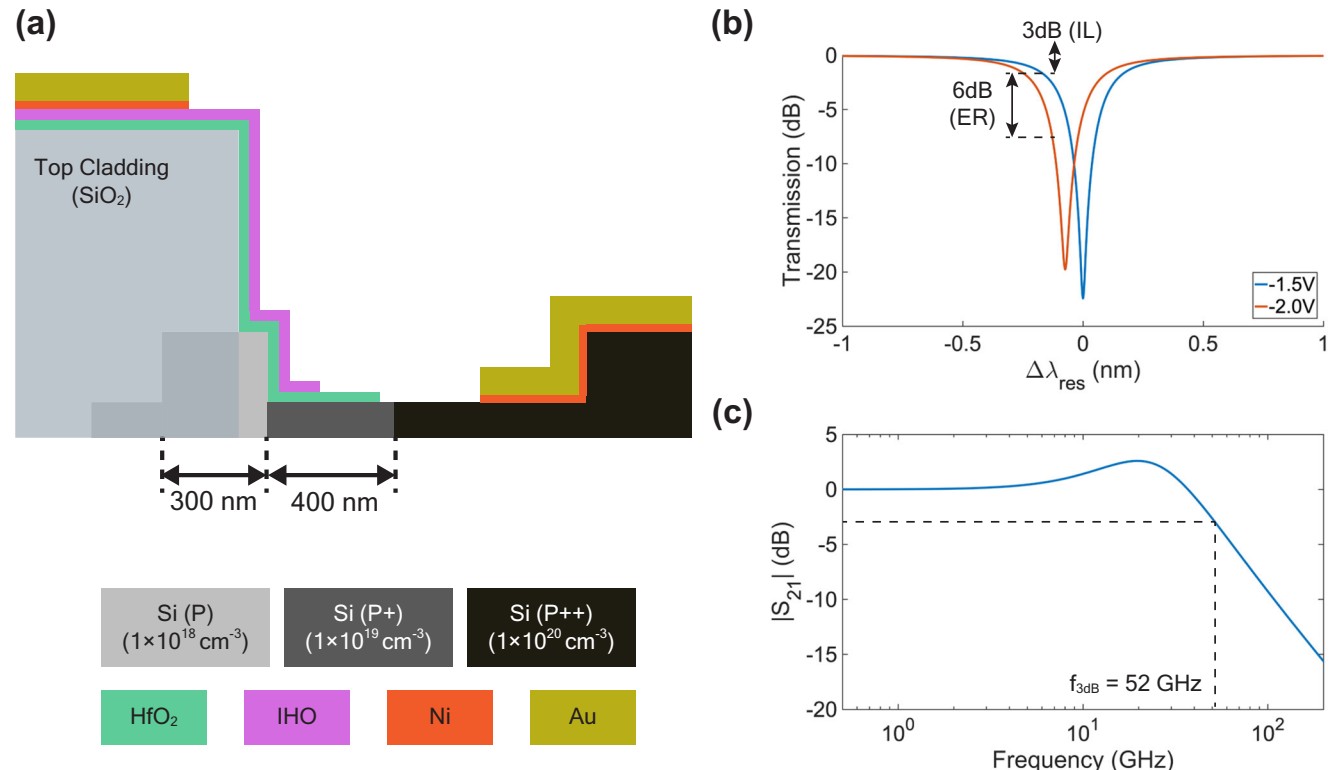

**Fig. 7 | Device optimization for improved performance. a** The cross-sectional view in the active region of the MRM with TCO only covering the outer sidewall of the ring. **b** Static simulation of the shifted spectra under −1.5 V and −2.0 V.

**c** Simulated E-O response ($S_{21}$) of the MRM with 52 GHz bandwidth. The E-O response simulation setup is shown in Supplementary Information VII.

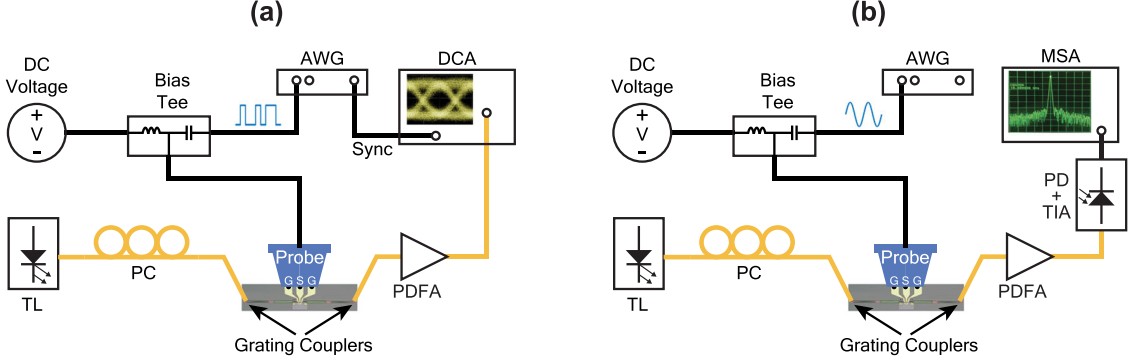

**Fig. 8 | Schematic of high-speed testing setups. a** Eye diagrams testing. **b** E-O response testing.

## Data availability

All data generated or analyzed during this study are included in this published article and its supplementary information files. The experimental data generated in this study are provided in the Source Data files. Source data are provided in this paper.

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

## Acknowledgements

This work is supported by the Intel URC project 76084461, the NSF GOALI project 2240352, the AFOSR MURI project FA9550-17-1-0071, and the NASA ESI program 80NSSC23K0195. Besides, we would like to thank the DURIP grant FA9550-20-1-0151 and Baylor University Mearse Endowment for equipment support of high-speed optoelectronics characterization, Materials Synthesis and Characterization Facility (MaSC) and the Electron Microscopy Facility at Oregon State University for our device fabrication. MaSC is part of the Northwest Nanotechnology Infrastructure, a National Nanotechnology Coordinated Infrastructure site at Oregon State University supported in part by the National Science Foundation (grant NNCI-1542101). We acknowledge Meer Sakib for the helpful discussion in design and characterization, Duanni Huang, James Jaussi, and Pavan Kumar Hanumolu for technical discussions.

## Author contributions

W.H. and A.X.W. were involved in the design, experiments, and manuscript writing. N.N. was involved in device fabrication. B.K. and J.F.C. were involved in atomic layer deposition. H.R. and R.K. were involved in the design. All authors reviewed the manuscript.

## Competing interests

The authors declare no competing interests.
