## [Peer Review File · Nature Communications]

REVIEWER COMMENTS

Reviewer #1 (Remarks to the Author):

The authors present the first high-speed optical modulator enabled by High Mobility Conductive Oxide that can reach several 10s Gb/s modulation when hybrid integration in a silicon photonics platform. The results are novel and exciting for the silicon photonics research community, which paves a potential route to continuously improve the energy efficiency of the silicon photonics transmitter efficiency in future.

Here I have some comments to improve the paper:

- (1) Lines 45-47. Can authors explain more about the ring modulator's critical aspects that can help reduce the CMOS driver power? Reducing the total power consumption for a combination of ring modulator and CMOS driver is more interesting. Is there any difference between a conventional PN ring modulator and a MOSCAP modulator for co-designing a CMOS driver?
- (2) "larger capacitance density" is used many times. The authors can explain a bit more. In some cases, a high capacitance density induces high carrier concentration, which helps increase the modulation efficiency. In some cases, it also increases the total capacitance and limits the bandwidth. The capacitance efficiency should be considered in the discussion.
- (3) When an arbitrary waveform generator (AWG) is used to pre-emphasize the signals, what are the key issues that cause the noise and bandwidth limit of the eye diagrams? More explanation can be added to the paper.
- (4) The RC bandwidth accounts for a source impedance of 50 ohms, resulting in an RC bandwidth of about 10 GHz instead of 40 GHz. The author should explain more about the simulation of Fig.7c and possibly an improved design allowing much lower capacitance.

Reviewer #2 (Remarks to the Author):

The article presents a modulator based on a ring resonator utilizing a transparent conductive oxide for efficient use of the plasma dispersion effect. The use of transparent conductive oxides to provide high efficiency modulation is a technique that has been reported in several research works as also evidenced by the different references in the manuscript. This work looks interesting, but I find it hard to judge whether the particular design utilized in this work is novel enough to merit publication in Nat. Comm. Compared to the other devices listed in the supplemental document I am not sure if the current device is a leap forward. The device in ref. 4 looks very similar in properties and I wonder if a small redesign of that device could operate also at a sub-volt driving. I also have several remarks concerning the work carried out.

- For ring resonators, it is known that their performance is affected strongly by fabrication variations and tolerances. Often several rings are put on a chip and the one with the best properties is taken. This work only reports on a single device, but surely other devices will have been characterized. How are these other devices performing? Somehow, the authors should give insight into the reproducibility of the technology. Was only one ring working?
- Ring resonators are very interesting to demonstrate an EO technology, but these devices are not particularly interesting for real applications. They are only working well at a distinct wavelength, prone to fabrication errors, wavelength must be accurately controlled, etc. Broadband devices typically need longer sections and there not the VpiL is the main figure of merit, also the Vpi x L x alpha (see for example the article by Azadeh DOI: 10.1364/Oe.23.023526). The authors must provide an analysis of the loss (in dB/cm) for the TCO coated section. Such loss values can be extracted from the ring's transmission profiles, and again, data from different rings will probably be essential to get a good estimate of the loss. How would a full-pi modulator section compare to other reported technologies?

- A remark on the text: at the end of the introduction (lines 62 to 73) the properties of the device are listed. This does not belong in an introduction and the numbers are already mentioned in the abstract and the conclusions. It is not good practice to repeat this once more in the introduction. Instead, I advise the authors to focus a bit more on how their technique is different from other reported works.

Reviewer #3 (Remarks to the Author):

The authors show a novel ring resonator based transparent conductive oxide (TCO) modulator. The device is energy efficient, can be operated under a low peak to peak drive voltage and shows significant bandwidth compared to other TCO based modulators in literature. The document is overall well written and the data seems to support the claims. However, the authors put in my opinion too much weight onto the low peak to peak drive voltage compared to the bandwidth of the device. In the broad context of silicon photonic modulators their device is less impressive (see for example DOI:10.1038/ncomms5008) while it seems a welcome and significant advancement in the closer context of TCO modulators [DOI:10.1063/5.0048712]. In this context the paper seems to really shine, it might be the first TCO modulator with >10 GHz EO bandwidth and is overall competitive with modulators shown in literature with some possible improvements outlined for future iterations of the device.

Beside this I have the following comments and suggestions to improve the overall manuscript.

1. Line 14-16, and line 39-40: please review the literature, there are silicon photonics modulators with the characteristics. They might not be the most common but they have been experimentally demonstrated, especially "An ultralow power athermal silicon modulator" published in 2014. Other references might also be appropriate here as an example M.T. Wade and Ayar labs have some work based on low voltage swing ring resonator modulators.
2. Line 22: I understand what you meant with parasitic capacitance only after reading the supplement.
3. Line 43-44: When the supply is limited to 1V using a voltage between 1.5 V and 2.3 V needs at least an integrated bias-Tee and an additional voltage supply which might be a case against your device.
4. Line 52-53: Please see the first comment, they have been reported a long time ago.
5. Line 109-111: Can you give the optical modulation amplitude (OMA) for this modulator? How did you optimize the tradeoff between insertion loss and extinction ratio?
6. Line 148: You choose an insertion loss of 3dB but it does not become clear why. Is this the point that gives you the optimal OMA, ER or is the choice arbitrary?
7. Line 164: What was used as the fitting curve? A model of the ring resonator taking detuning, peaking and RC bandwidth into account I assume?
8. Line 175: The different PRBS length imply there was an issue at low speeds is this assumption true?
9. Line 176-177: The capacitance is part of your device. In my opinion this discussion should be moved to the main document, since otherwise people might assume you are understating your capacitance.
10. Line 205-206: In my opinion it would be better to stay with ITiO here to stick closer to the experimental results at hand.
11. Line 209: Is there a concrete system you envisioned such that you optimize to 3dB IL and > 6dB ER?
12. Fig. 7: You seem to optimize the device geometry here. I don't understand why you keep the p doping on the left side of the resonator since this will increase your losses and your capacitor is only on the right side. Since you discussed bending losses in the supplement, maybe it would be interesting to invert the bending orientation, such that the ITiO side is at the outer side wall. While this might reduce the efficiency it drastically reduces the bending losses even for smaller resonators. An alternative would be to etch sub-wavelength sections in to the slab as theoretically investigated by me in "Sub-wavelength tunneling barrier in rib waveguide microring modulators

with vanishing bending losses”.

13. Line 224-226: How would this performance be of ITiO would be used with the optimized structure?

Baylor University

November 1st, 2023

Dear Reviewers,

We would like to thank you for your efforts in reviewing our manuscript entitled "Sub-Volt High-Speed Silicon MOSCAP Microring Modulator Driven by High Mobility Conductive Oxide." In this response letter, please find answers to your comments and the summary of the revision according to your suggestions. The changes are also highlighted in the resubmitted manuscript.

Sincerely,

Professor and Mearse Endowed Chair
Department of Electrical and Computer Engineering
Baylor University
One Bear Place #97356, Waco, TX, 76798-7356
Email: alan_wang@baylor.edu
Office: 254-710-6859
https://sites.baylor.edu/alan_wang

Reviewer 1:

Comment1.

Lines 45-47. Can authors explain more about the ring modulator's critical aspects that can help reduce the CMOS driver power? Reducing the total power consumption for a combination of ring modulator and CMOS driver is more interesting. Is there any difference between a conventional PN ring modulator and a MOSCAP modulator for co-designing a CMOS driver?

A: We appreciate the Reviewer for bringing up this insightful discussion. According to real product development experiences from our Intel co-authors, the energy consumption of optical modulator is insignificant compared with that of its CMOS driver¹. High-voltage modulator drivers come with high energy cost because of the following factors: Firstly, a substantial amount of energy is expended in charging the capacitor to a higher voltage; Secondly, the driver necessitates cascoding to enable operation beyond the supply voltage limits, resulting in a bandwidth penalty that requires higher power consumption to compensate for the lost bandwidth. Lastly, it significantly increases the complexity of the driver circuitry.

Therefore, reducing the driving voltage of the modulator not only lowers the power consumption of the modulator itself, but also enables energy-efficient, high-speed modulator drivers using advanced CMOS nodes which cannot support the high driving voltage of existing silicon photonic modulators². For example, Ref [1] shows that 3V_{pp} CMOS driver consumed 6pJ/bit while a 1.55V_{pp} driver only requires 0.685pJ/bit.

Typically, conventional PN ring modulators require a higher driver voltage due to the weak plasma dispersion effect of silicon and the small capacitance density. In contrast, MOSCAP ring modulators offer higher capacitance density and better modulation efficiency using heterogeneously integrated gate material, resulting in reduced driving voltage that can benefit the design and energy efficiency of CMOS drivers. A brief explanation is added into the revised manuscript.

Comment2.

"larger capacitance density" is used many times. The authors can explain a bit more. In some cases, a high capacitance density induces high carrier concentration, which helps increase the modulation efficiency. In some cases, it also increases the total capacitance and limits the bandwidth. The capacitance efficiency should be considered in the discussion.

A: We fully agree with the Reviewer that higher capacitance density can improve the E-O modulation efficiency but also negatively affect the modulation bandwidth. Therefore, optimal capacitance density with balanced performance is a crucial aspect of the MOSCAP microring modulator design. We have included more discussion in the revised manuscript.

Comment3.

When an arbitrary waveform generator (AWG) is used to pre-emphasize the signals, what are the key issues that cause the noise and bandwidth limit of the eye diagrams? More explanation can be added to the paper.

A: We appreciate the Reviewer's suggestion. In the revised manuscript, additional discussion is included. Pre-emphasis can lead to a cleaner and more open eye diagram. However, it's important to note that pre-emphasis doesn't increase the device's bandwidth. At higher data rates, the primary limiting factor for the eye diagram is the actual bandwidth of the device, which is the RC bandwidth for our current device. Additionally, in our testing configuration, the use of an optical amplifier introduced amplified spontaneous emission (ASE) noise, which had a detrimental effect on the clarity of the eye diagram. However, the ASE noise can be filtered out by a narrow band-pass optical filter, which will be purchased for our future research.

Comment4.

The RC bandwidth accounts for a source impedance of 50 ohms, resulting in an RC bandwidth of about 10 GHz instead of 40 GHz. The author should explain more about the simulation of Fig.7c and possibly an improved design allowing much lower capacitance.

A: The RC bandwidth of the device is only determined by its own physics characteristics, such as the capacitance and series resistance in the modulator. The 50Ω source impedance is essential to impedance matching with the RF cable to deliver the RF power to the device. However, due to the ultra-short electrode length of the microring modulator, we can simplify the device to a lumped element and as results, the 50Ω source impedance will not affect the simulation and measurement of the bandwidth. A brief explanation is added.

Reviewer 2:

Comment1.

Compared to the other devices listed in the supplemental document I am not sure if the current device is a leap forward. The device in ref. 4 looks very similar in properties and I wonder if a small redesign of that device could operate also at a sub-volt driving.

A: We included a holistic comparison with other silicon microring modulators in Table.1 of the Supplementary Information. The performance of these devices always needs the balance among driving voltage, insertion loss, and bandwidth. Some modulators did achieve sub-volt driving voltage through sacrifice of other performance. We cannot comment on ref.4 to address the Reviewer's question because we do not have all detailed design parameters needed. However, we must point out that heterogeneous integration with more efficient semiconductor materials is critical to further improve the performance matrix compared with pure silicon photonic devices such as ref. 4.

Comment2.

For ring resonators, it is known that their performance is affected strongly by fabrication variations and tolerances. Often several rings are put on a chip and the one with the best properties is taken. This work only reports on a single device, but surely other devices will have been characterized. How are these other devices performing? Somehow, the authors should give insight into the reproducibility of the technology. Was only one ring working?

A: We appreciate the Reviewer's point regarding the reproducibility of our devices. Indeed, hundreds of silicon microrings were fabricated on the wafer by Intel's fab. However, we also used post-foundry fabrication processes involving RF-sputtering and e-beam lithography to pattern the ITiO gates and metal electrodes. As results, we were able to fabricate only eight fully completed E-O modulators. Each of these devices was functional, but they exhibited variations in their performance due to different waveguide gaps for optimal coupling in our design. Devices with coupling conditions close to the critical coupling exhibited deep resonant dips, allowing for sub-volt modulation. On the other hand, devices away from the critical coupling had shallower resonant dips, requiring higher driving voltages. We have incorporated a discussion on the critical coupling condition in the revised manuscript. We also included two additional modulator results as examples in the Supplementary Information. These examples demonstrated the reproducibility of our fabrication process and emphasized how the critical coupling condition impacts device performance.

Comment3.

Ring resonators are very interesting to demonstrate an EO technology, but these devices are not particularly interesting for real applications. They are only working well at a distinct wavelength, prone to fabrication errors, wavelength must be accurately controlled, etc. Broadband devices typically need longer sections and there not the $V_{\pi}L$ is the main figure of merit, also the $V_{\pi} \times L \times \alpha$ (see for example the article by Azadeh DOI: 10.1364/Oe.23.023526). The authors must provide an analysis of the loss (in dB/cm) for the TCO coated section. Such loss values can be extracted from the ring's transmission profiles, and again, data from different rings will probably be essential to get a good estimate of the loss. How would a full- π modulator section compare to other reported technologies?

A: We agree with the Reviewer that microring modulators face many technical challenges for real optical communication systems and great amount of efforts have been dedicated to perfect the technology in the past decade. Successful examples include a 1.6Tb/s photonics engine for data centers developed by Intel using 16 microring modulator array with thermal heaters³.

As suggested by the Reviewer, we included an analysis of the loss (in dB/cm) for the TCO-coated section. Additionally, we added a section for comparing $V_{\pi} \times L \times \alpha$ with other reported modulators in the Supplementary Information. Before applying the ITiO coating, the passive microring waveguide exhibited a loss (α) of 24 dB/cm. After the complete device fabrication, the ITiO-coated silicon waveguide showed an α of 138 dB/cm. We did not compare the V_{π} because it is inversely proportional to the electrode length. Instead, $V_{\pi} \times L$ is widely adopted for performance evaluation, which is already included in Table.1 of the Supplementary Information. Compared to other reported technologies, our device shows superior performance in both O-band and C-band with detailed comparison in the Supplementary Information.

Comment4.

A remark on the text: at the end of the introduction (lines 62 to 73) the properties of the device are listed. This does not belong in an introduction and the numbers are already mentioned in the abstract and the conclusions. It is not good practice to repeat this once more in the introduction. Instead, I advise the authors to focus a bit more on how their technique is different from other reported works.

A: We appreciate the Reviewer's suggestions on the manuscript writing. We revised the introduction to avoid the repetition of device properties and have focused on highlighting the uniqueness of our device in the context of TCO modulators.

Reviewer 3:

The authors put in my opinion too much weight onto the low peak to peak drive voltage compared to the bandwidth of the device. In the broad context of silicon photonic modulators their device is less impressive (see for example DOI:10.1038/ncomms5008) while it seems a welcome and significant advancement in the closer context of TCO modulators [DOI:10.1063/5.0048712]. In this context the paper seems to really shine, it might be the first TCO modulator with >10 GHz EO bandwidth and is overall competitive with modulators shown in literature with some possible improvements outlined for future iterations of the device.

A: We fully agree with the Reviewer that high bandwidth is one of the most critical requirements for silicon modulators, which has been extensively investigated in past years. However, lowering the driving voltage is also of high interests to optical components companies such as Intel. Lowering the driving voltage brings exclusive advantages to CMOS drivers and the overall system performance as discussed in response to Reviewer 1 Comment 1^{1,2}. Therefore, our primary focus in this work was to reduce the driving voltage to pave the way for energy-efficient optical interconnects. As discussed in the manuscript, we do keep the bandwidth requirement in our working list and our next goal is to reduce the parasitic capacitance to achieve higher bandwidth.

We also thank the Reviewer for his/her appreciation of our work in the context of TCO modulators. We highlight that our work represents a major advancement of first TCO modulator capable of high-speed (25Gb/s) clear-eye E-O modulation, which has been added into the Introduction. Additionally, we appreciate the Reviewer for suggesting more reference articles covering low driving voltage modulators, which have been included in the revised manuscript. Discussions can be found in our response to Comment 1.

Comment1.

Line 14-16, and line 39-40: please review the literature, there are silicon photonics modulators with the characteristics. They might not be the most common but they have been experimentally demonstrated, especially "An ultralow power athermal silicon modulator" published in 2014. Other references might also be appropriate here as an example M.T. Wade and Ayar labs have some work based on low voltage swing ring resonator modulators.

A: We appreciate the Reviewer's suggestion of additional articles on low driving voltage silicon modulators, which are now incorporated in the revised manuscript for a more comprehensive comparison.

"An ultralow power athermal silicon modulator" is indeed an impressive demonstration of low-voltage silicon modulator, but on a microdisk platform⁴. Microdisk has intrinsic advantages over microring due to its smaller volume and better overlap with free carrier modulation. But it did come with a drawback of large leakage current. Another noteworthy publication by M.T. Wade and Ayar

Labs also achieved a low driving voltage of $1 V_{pp}$ ⁵. It also utilizes an ultra-compact ring to improve the E-O modulation efficiency, but with limited RC bandwidth of 2 GHz.

Our work presented a different approach to achieving low driving voltage through heterogeneously integration with a microring modulator. As a relatively new approach with significant challenges in hybrid material integration, we acknowledge that there is huge room for future improvement.

Comment2.

Line 22: I understand what you meant with parasitic capacitance only after reading the supplement.

A: We apologize for the confusion. The concept of the parasitic capacitance was not clear in the initial submission. We have revised the abstract to refer to 'RC bandwidth' and included additional explanations of parasitic capacitance in the main manuscript to reduce the confusion to readers.

Comment3.

Line 43-44: When the supply is limited to 1V using a voltage between 1.5 V and 2.3 V needs at least an integrated bias-Tee and an additional voltage supply which might be a case against your device.

A: This is indeed the case in our device testing. However, in real optical transmitters consisting of silicon modulators with co-packaged CMOS drivers⁶, both the DC bias and AC swing will be supplied by the same CMOS driver without the need of bias-Tee.

Comment4.

Line 52-53: Please see the first comment, they have been reported a long time ago.

A: We agree with the Reviewer and revised the writing: "Nevertheless, there is still a strong desire to develop high-speed Si-MRMs with sub-volt V_{pp} and large E-O modulation efficiency".

Comment5.

Line 109-111: Can you give the optical modulation amplitude (OMA) for this modulator? How did you optimize the tradeoff between insertion loss and extinction ratio?

A: We have added more description to the manuscript. The OMA is 0.024 mW (assuming 0dBm optical input power with 12dB loss from grating couplers). However, it's worth noting that OMA can vary depending on the input optical power and device loss. To provide a fair comparison of the device's performance, we primarily focused on the extinction ratio (ER) and insertion loss (IL). We chose a 3 dB IL point based on the balanced consideration of output power and driving voltage. If we have pursued a 6 dB IL, the driving voltage will be even lower. Notably, the combination of 3dB IL and 6dB ER is also recommended in other literatures⁷.

Comment6.

Line 148: You choose an insertion loss of 3dB but it does not become clear why. Is this the point that gives you the optimal OMA, ER or is the choice arbitrary?

A: The rationale for our choice was explained in the answer to Comment 5. A brief explanation is added in the revised manuscript for clarity.

Comment7.

Line 164: What was used as the fitting curve? A model of the ring resonator taking detuning, peaking and RC bandwidth into account I assume?

A: Yes, the fitting curve is formed by fitting the discrete data points of the measured RF power to show the device's EO bandwidth, which considers the detuning, peaking, and RC bandwidth. We added more explanation to the revised manuscript.

Comment8.

Line 175: The different PRBS length imply there was an issue at low speeds is this assumption true?

A: Yes, this is a practical issue related to high-speed testing. Longer PRBS sequences are necessary to provide higher resolution for characterizing higher data rates. We have added more descriptions to the manuscript.

Comment9.

Line 176-177: The capacitance is part of your device. In my opinion this discussion should be moved to the main document, since otherwise people might assume you are understating your capacitance.

A: We appreciate the Reviewer's feedback. Since the capacitance measurement is a fundamental characterization, we feel it is more appropriate to present it in the Supplementary Information. However, we did include additional details in the main document to provide a comprehensive description of our device, thereby avoiding any potential misinterpretation.

Comment10.

Line 205-206: In my opinion it would be better to stay with ITiO here to stick closer to the experimental results at hand.

A: We fully agree with the Reviewer about future research approaches. Our modulator research with ITiO is still our top priority and focus. Concurrently, we are exploring IHO, which can potentially offer higher mobility for better performance.

Comment11.

Line 209: Is there a concrete system you envisioned such that you optimize to 3dB IL and > 6dB ER?

A: As we discussed in Comment 5, there is a trade-off between IL and ER. We set our device measurement at 3 dB IL and 6 dB ER,⁷ which is a more rigorous standard to evaluate the driving voltage and energy efficiency. We could claim better results if we operate the device at higher IL.

Comment12.

Fig. 7: You seem to optimize the device geometry here. I don't understand why you keep the p doping on the left side of the resonator since this will increase your losses and your capacitor is only on the right side. Since you discussed bending losses in the supplement, maybe it would be interesting to invert the bending orientation, such that the ITiO side is at the outer side wall. While this might reduce the efficiency it drastically reduces the bending losses even for smaller resonators. An alternative would be to etch sub-wavelength sections into the slab as theoretically investigated by me in "Sub-wavelength tunneling barrier in rib waveguide microring modulators with vanishing bending losses".

A: We appreciate the reviewer for prompting the insightful discussions and providing valuable suggestions. Regarding the TCO-gated MOSCAP Si-MRM, it's important to note that even with the light p-doping on the Si waveguide, the loss reduction is not significant because the dominant loss comes from the TCO layer. Concerning the bending orientation, we fully agree that inverting the bending orientation can reduce bending losses. However, the optimized structure is designed with the TCO layer primarily covering the outer side wall. The structure with an inverted bending orientation would cause a longer path for the Si to reach the capacitor at the outer sidewall. It would introduce larger series resistance and reduce the bandwidth, which is undesirable for our specific design.

Regarding the sub-wavelength structure proposed by the reviewer, while it is an interesting idea, it doesn't align with the design requirements of Intel's fab. However, it warrants future investigations using our own fabrication processes. A brief plan is added in the revised manuscript.

Comment13.

Line 224-226: How would this performance be of ITiO would be used with the optimized structure?

A: When the optimized structure employs ITiO, it can still achieve an E-O efficiency comparable to the device using IHO. However, it's essential to note that ITiO exhibits lower carrier mobility. As a result, the ITiO-based device may exhibit a lower Q-factor, which will require a higher driving voltage. Additionally, due to the reduced carrier mobility of ITiO, the device's RC bandwidth is also reduced. Nonetheless, it's worth noting that our current ITiO mobility of $62 \text{ cm}^2/(\text{V}\cdot\text{s})$ is not yet optimal. In a literature⁸, ITiO can achieve the mobility up to $105 \text{ cm}^2/(\text{V}\cdot\text{s})$. In such a case, an optimal design incorporating ITiO can outperform our current device significantly.

Reference:

1. Li, H. *et al.* A 3-D-Integrated Silicon Photonic Microring-Based 112-Gb/s PAM-4 Transmitter With Nonlinear Equalization and Thermal Control. *IEEE J. Solid-State Circuits* **56**, 19–29 (2021).
2. Levy, C. *et al.* A 3D-integrated $8\lambda \times 32$ Gbps λ Silicon Photonic Microring-based DWDM Transmitter. in *2023 IEEE Custom Integrated Circuits Conference (CICC)* 1–2 (2023).
3. Fatholouloumi, S. *et al.* 1.6 Tbps Silicon Photonics Integrated Circuit and 800 Gbps Photonic Engine for Switch Co-Packaging Demonstration. *J. Light. Technol.* **39**, 1155–1161 (2021).
4. Timurdogan, E. *et al.* An ultralow power athermal silicon modulator. *Nat. Commun.* **5**, 4008 (2014).
5. Gevorgyan, H., Khilo, A., Wade, M. T., Stojanović, V. M. & Popović, M. A. Miniature, highly sensitive MOSCAP ring modulators in co-optimized electronic-photonics CMOS. *Photon. Res.* **10**, A1–A7 (2022).
6. Li, H. *et al.* A 106 Gb/s 2.5 Vppd Linear Microring Modulator Driver with Integrated Photocurrent Sensor in 28nm CMOS. in *OFC M2D.1* (2022).
7. Sun, C. *et al.* Single-chip microprocessor that communicates directly using light. *Nature* **528**, 534–538 (2015).
8. Hashimoto, R., Abe, Y. & Nakada, T. High Mobility Titanium-Doped In₂O₃ Thin Films Prepared by Sputtering/Post-Annealing Technique. *Appl. Phys. Express* **1**, 15002 (2008).

REVIEWER COMMENTS

Reviewer #1 (Remarks to the Author):

In reply to Reviewer 1 comment 4, the authors neglect the driver source resistance/impedance. For the lumped model, the source resistance/impedance plays a key role and affects the total device modulation Bandwidth. Therefore, I can't believe the estimation of the optimized device will have a high bandwidth >50 GHz. The Intel co-author's silicon ring modulator papers have already shown that ring intrinsic bandwidth (excluding source resistance) is higher than actual device bandwidth (including source resistance). Additionally, leading authors can refer to the book, Chapter-6, Semiconductor Devices for High-Speed Optoelectronics, GIOVANNI GHIONE.

I expect the authors to modify the part of the optimized device and explain the capacitance and resistance change in different regions before and after optimization. A bandwidth simulation model can be included in the supplementary file.

Reviewer #2 (Remarks to the Author):

The authors have adequately answered my questions and concerns, they made the necessary changes in the main manuscript and they added important extra data in the supplementary document.

Reviewer #3 (Remarks to the Author):

The authors have adequately addressed most of the comments. Particularly, the broader context of other silicon photonics modulators is more comprehensively represented. The rationale for the chosen detuning leading to the tradeoff between insertion loss and extinction ratio has been appropriately explained.

I have the following specific comments:

- Lines 201 to 204 seem redundant in light of the newly added lines 194 to 201. Please remove the previous sentences as they duplicate the newer content.
- In the review attachments, I included a sketch (Image.ppt) to illustrate the suggestion I made in Reviewer3 Comment 12. This sketch demonstrates the potential effect of inverting the bending orientation, indicating that relocating the capacitor to the inner side wall would not significantly increase the path length from the contact in Si to reach the capacitor and series resistance. The initial assessment suggests that placing the capacitor on the inner side wall could reduce the phase shift efficiency by approximately 20%. It's plausible that fine-tuning other waveguide dimensions might partly recover this reduction. Additionally, this adjustment could enable a more compact bend resonator. As a counterbalance to the reduced circumference, a consideration could be made to extend the modulation region to cover 100% of the ring instead of the current 70%. However, to establish contact with the central electrode, the formation of vias would be necessary. This suggestion is intended for a potential design study and is not expected to be immediately reflected in the current manuscript. My intent was solely to clarify the idea and provide a visual representation for future exploration.

Baylor University

December 3rd, 2023

Dear Reviewers,

We would like to thank you for your efforts in reviewing our manuscript entitled "Sub-Volt High-Speed Silicon MOSCAP Microring Modulator Driven by High Mobility Conductive Oxide." In this response letter, please find answers to your comments and the summary of the revision according to your suggestions. The changes are also highlighted in the resubmitted manuscript.

Sincerely,

Professor and Mearse Endowed Chair
Department of Electrical and Computer Engineering
Baylor University
One Bear Place #97356, Waco, TX, 76798-7356
Email: alan_wang@baylor.edu
Office: 254-710-6859
https://sites.baylor.edu/alan_wang

Reviewer 1:

Comment 1.

In reply to Reviewer 1 comment 4, the authors neglect the driver source resistance/impedance. For the lumped model, the source resistance/impedance plays a key role and affects the total device modulation Bandwidth. Therefore, I can't believe the estimation of the optimized device will have a high bandwidth >50 GHz. The Intel co-author's silicon ring modulator papers have already shown that ring intrinsic bandwidth (excluding source resistance) is higher than actual device bandwidth (including source resistance). Additionally, leading authors can refer to the book, Chapter-6, Semiconductor Devices for High-Speed Optoelectronics, GIOVANNI GHIONE.

I expect the authors to modify the part of the optimized device and explain the capacitance and resistance change in different regions before and after optimization. A bandwidth simulation model can be included in the supplementary file.

A: We appreciate the reviewer's insightful comments and the provided references. We fully agree that the source impedance affects the modulation bandwidth of the entire optical transmitter system. However, this manuscript focuses on evaluating the performance of the microring modulator (MRM) itself and comparing it with other MRMs. In line with many other literature practices¹⁻⁵, only the intrinsic bandwidth of the MRM is discussed in this manuscript, excluding the source impedance, which is de-embedded during the characterization. The reviewer's suggestion of including the source impedance will definitely be valid for research on optical transmitters including both modulators and driver circuits.

Regarding the Intel co-author's silicon ring modulator papers^{4,5}, it is true that the ring achieved a high RC bandwidth, excluding source impedance. However, the actual ring bandwidth is lower majorly due to the photon-lifetime-limited optical bandwidth, not because of the source impedance. In the case of the MRM, the bandwidth is determined by both the photon-lifetime-limited optical and RC electrical bandwidths ($1/f_{3dB}^2 = 1/f_{OP}^2 + 1/f_{RC}^2$)⁶. Again, the source impedance is excluded for the characterization of MRM itself.

For further discussion with the Reviewer about the modulation bandwidth of the entire optical transmitter, the driver is usually carefully designed to achieve a low output impedance. This design choice ensures that the system's modulation bandwidth is primarily determined by the MRM itself, with minimal impact from the source impedance. For instance, the MRM driven by a low output impedance of 4 Ω (typical MRM itself has a series resistance $>30\Omega$) has been successfully demonstrated⁷.

Our experiment has presented the device's pre-optimized results. Therefore, the proposed optimization strategy is introduced to illustrate the potential bandwidth improvement through simulation. We revised the manuscript by stating that the RC bandwidth excludes source impedance, which would be more clear according to the discussion with the Reviewer. The bandwidth simulation model adheres to the guidelines provided on the Ansys/Optics website, which was included in the Supplementary Information.

Reviewer 2:

The authors have adequately answered my questions and concerns, they made the necessary changes in the main manuscript, and they added important extra data in the supplementary document.

A: We appreciate the Reviewer's thorough review of our answers and revised manuscript.

Reviewer 3:

Comment1.

Lines 201 to 204 seem redundant in light of the newly added lines 194 to 201. Please remove the previous sentences as they duplicate the newer content.

A: We appreciate the Reviewer's valuable comment in refining the content. We have deleted the redundant content.

Comment2.

In the review attachments, I included a sketch (Image.ppt) to illustrate the suggestion I made in Reviewer3 Comment 12. This sketch demonstrates the potential effect of inverting the bending orientation, indicating that relocating the capacitor to the inner side wall would not significantly increase the path length from the contact in Si to reach the capacitor and series resistance. The initial assessment suggests that placing the capacitor on the inner side wall could reduce the phase shift efficiency by approximately 20%. It's plausible that fine-tuning other waveguide dimensions might partly recover this reduction. Additionally, this adjustment could enable a more compact bend resonator. As a counterbalance to the reduced circumference, a consideration could be made to extend the modulation region to cover 100% of the ring instead of the current 70%. However, to establish contact with the central electrode, the formation of vias would be necessary. This suggestion is intended for a potential design study and is not expected to be immediately reflected in the current manuscript. My intent was solely to clarify the idea and provide a visual representation for future exploration.

A: We greatly appreciate the Reviewer's time taken to provide the illustration and suggestion. The proposed idea is certainly worthy of exploration in the future.

Reference:

1. Zhang, W. *et al.* Harnessing plasma absorption in silicon MOS ring modulators. *Nat. Photonics* **17**, 273–279 (2023).
2. Chan, D. W. U. *et al.* Ultra-Wide Free-Spectral-Range Silicon Microring Modulator for High Capacity WDM. *J. Light. Technol.* **40**, 7848–7855 (2022).
3. Cai, H., Fu, S., Yu, Y. & Zhang, X. Lateral-Zigzag PN Junction Enabled High-Efficiency Silicon Micro-Ring Modulator Working at 100Gb/s. *IEEE Photonics Technol. Lett.* **34**, 525–528 (2022).
4. Sakib, M. *et al.* A 240 Gb/s PAM4 Silicon Micro-Ring Optical Modulator. in *2022 Optical Fiber Communications Conference and Exhibition (OFC)* 1–3 (2022).
5. Sun, J. *et al.* A 128 Gb/s PAM4 Silicon Microring Modulator With Integrated Thermo-Optic Resonance Tuning. *J. Light. Technol.* **37**, 110–115 (2019).
6. Dong, P. *et al.* Low V_{pp}, ultralow-energy, compact, high-speed silicon electro-optic modulator. *Opt. Express* **17**, 22484–22490 (2009).
7. Moscoso-Mártir, A. *et al.* Silicon Photonics Transmitter with SOA and Semiconductor Mode-Locked Laser. *Sci. Rep.* **7**, 13857 (2017).

REVIEWERS' COMMENTS

Reviewer #1 (Remarks to the Author):

Authors now make it clear about their bandwidth claim.